# Enhancing security in financial transactions: a novel blockchain-based federated learning framework for detecting counterfeit data in fintech

Hasnain Rabbani[1], Muhammad Farrukh Shahid[1], Tariq Jamil Saifullah Khanzada[2,3], Shahbaz Siddiqui[1], Mona Mamdouh Jamjoom[4], Rehab Bahaaddin Ashari[3], Zahid Ullah[3], Muhammad Umair Mukati[5] and Mustafa Nooruddin[6]

[1] Computer Science, FAST School of Computing, FAST-NUCES, Karachi, Sindh, Pakistan
[2] Computer Systems Engineering Department, Mehran UET, Hyderabad, Sindh, Pakistan
[3] Department of Information Systems, King Abdulaziz University, Jeddah, Saudi Arabia
[4] Department of Computer Sciences, College of Computer and Information Sciences, Princess Nourah bint Abdulrahman University, Riyadh, Saudi Arabia
[5] Department of Electrical and Photonics Engineering, Technical University of Denmark, Denmark, DTU, Denmark
[6] College of Engineering, Karachi Institute of Economics and Technology, Karachi, Sindh, Pakistan



Corresponding author
Muhammad Farrukh Shahid,
mfarrukh.shahid@nu.edu.pk

## ABSTRACT

Fintech is an industry that uses technology to enhance and automate financial services. Fintech firms use software, mobile apps, and digital technologies to provide financial services that are faster, more efficient, and more accessible than those provided by traditional banks and financial institutions. Fintech companies take care of processes such as lending, payment processing, personal finance, and insurance, among other financial services. A data breach refers to a security liability when unapproved individuals gain access to or pilfer susceptible data. Data breaches pose a significant financial, reputational, and legal liability for companies. In 2017, Equifax suffered a data breach that revealed the personal information of over 143 million customers. Combining federated learning (FL) and blockchain can provide financial institutions with additional insurance and safeguards. Blockchain technology can provide a transparent and secure platform for FL, allowing financial institutions to collaborate on machine learning (ML) models while maintaining the confidentiality and integrity of their data. Utilizing blockchain technology, FL can provide an immutable and auditable record of all transactions and data exchanges. This can ensure that all parties adhere to the protocols and standards agreed upon for data sharing and collaboration. We propose the implementation of an FL framework that uses multiple ML models to protect consumers against fraudulent transactions through blockchain. The framework is intended to preserve customer privacy because it does not mandate the exchange of private customer data between participating institutions. Each bank trains its local models using data from its consumers, which are then combined on a centralised federated server to produce a unified global model. Data is neither stored nor exchanged between institutions, while models are trained on each institution's data.

## INTRODUCTION

In today's modern era, financial institutions have progressively expanded their services to the public through Internet banking. The use of electronic payment methods has become crucial in the highly competitive financial landscape, enabling convenient purchases of goods and services (*Bin Sulaiman, Schetinin & Sant, 2022*). Fintech is a term used to describe the use of technology to improve and automate financial services. Fintech companies use technology to provide financial services in a more efficient, convenient, and affordable way than traditional financial institutions (*Barbu et al., 2021*). Figure 1 shows the conceptual view of the fintech industry (*VectorStock, 2007*). Fintech companies offer customers the convenience of cards as an alternative to cash for making purchases. Credit cards provide consumers with various benefits, including purchase protection, which safeguards them against issues such as damaged, lost, or stolen goods (*Weichert, 2017*). In 2022, Mastercard issued 1,023 million cards in the first quarter of the year, 1,045 million cards in the second quarter of the year, 1,061 million cards in the third quarter, and 1,034 million cards in the fourth quarter of the year. This data shows that the number of Mastercard credit cards issued has consistently increased throughout the year, indicating that these credit cards are becoming increasingly popular among consumers (*Zen, 1966*). According to the Nilson report (*Nilson Report, 2024*), as of December 31, 2022, the total number of Visa and Mastercard credit, debit, and prepaid cards in circulation amounted to 1.91 billion. These statistics show that card-based transactions have become popular and convenient for end-users. Customers heavily rely on the convenience provided by fintech to pay bills, make merchant payments through point-of-sale (POS) machines, and transfer money using various digital platforms such as Interbank Fund Transfer (IFT), Instant Bank Fund Transfer (IBFT), digital wallets, and credit and debit cards. With all these benefits for customers, there has been a significant surge in credit card counterfeit activities. Credit cards are an attractive target for fraudsters since a substantial amount of money can be obtained quickly and relatively easily with low risk (*Kalmykova & Ryabova, 2016*). There are multiple types of credit card fraud, including online fraud, offline fraud, application fraud, and counterfeit. Online counterfeiting can occur through web transactions, phone shopping, or when the credit card owner is not present. On the other hand, offline counterfeiting involves criminals using stolen plastic cards in stores (*Ng & Kwok, 2017*). Application counterfeit is a serious type of counterfeit where false or stolen personal information is used to acquire a credit card with no intention of repayment. Counterfeit involves the unauthorized use of credit card details for remote transactions (*Yang et al., 2019*). Fraudsters use a variety of methods, such as phishing, skimming, and identity theft, to commit these crimes, but the goal is always the same: to steal money or personal information. These types of cyberattacks can be costly for banks as well as for customers. Customers can lose their hard-earned money, and banks can lose the trust of

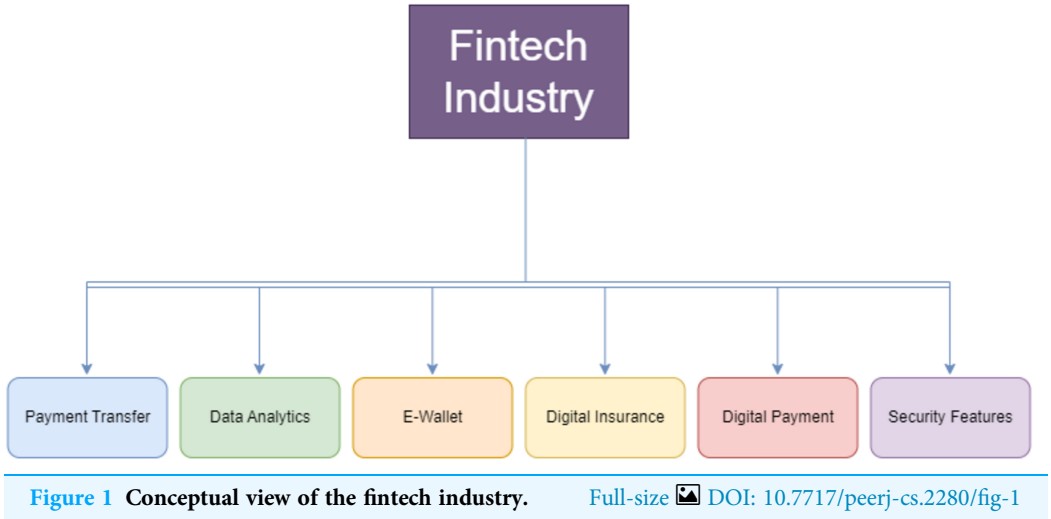

**Figure 1 Conceptual view of the fintech industry.**

the customer due to a data breach. To mitigate data breaches, banks should consider implementing AI-based systems that can effectively detect and identify such attempts (*Lacruz & Saniie, 2021*). Many of the banks nowadays implement ML-based models to prevent this type of fraud. To enable the detection of fraudulent transactions, ML models rely on training data. Banks, with their widespread branch networks, can leverage this advantage to facilitate customers' needs. Each branch can have its customer information. This information (dataset) needs to be in a centralized environment for training. Here comes the threat of data breaches and the exposure of customer personal information. To enhance security and privacy and mitigate the risk of data breaches, an additional layer of security and privacy can be implemented through the incorporation of federated learning (FL). As a privacy-preserving technology, FL is an artificial intelligence (AI) model that can accelerate the financing process using collaboration and communication (*Ashta & Herrmann, 2021*). FL doesn't expect information to be moved to a focal data set, which safeguards information protection and minimizes the risks of information security (*Cao, Yang & Yu, 2021*). The integration of AI and machine learning (ML) in fintech not only enhances efficiency but also allows for advanced data analysis, risk assessment, and personalized financial solutions, ultimately revolutionizing the industry's ability to adapt and cater to the diverse needs of individuals and businesses.

## Fintech with artificial intelligence

The fusion of AI and fintech has brought innovative solutions that empower customers and businesses. In this connection, fintech, together with AI, has facilitated significant positive changes, enabling financial institutions to make more informed decisions, streamline operations, and enhance customer experiences. Data science serves as the foundation that enables fintech to execute faster and more accurate decision-making processes compared to traditional institutions (*Guo & Polak, 2021*). AI-driven solutions are transforming industries, including risk assessment, customer service automation, and fraud detection. AI helps streamline processes, provide individualized financial advice, and ensure regulatory compliance by processing massive amounts of information in real time.

AI-powered trading algorithms and robo-advisors automate investing techniques (*Bayramoğlu, 2021*). The impact of AI extends beyond only increasing productivity; it also makes the market for both consumers and enterprises safer and more accessible.

Machine learning is a category of AI that enables software applications to improve their accuracy in predicting outcomes without being specifically engineered to do so. ML algorithms have influenced numerous sectors by improving customer service and growing client reach. Fintech is a dynamic and innovative field that fully benefits from developments in information and communication technology (*Stojanović et al., 2021*). ML is an effective technology for extracting insights from data and creating predictions. The security and integrity of consumer data play a significant role in fintech. Traditional security solutions are no longer suitable for preserving sensitive information given the rising number of digital transactions and the high degree of fraudulent activity. As a result, ML has emerged as an essential mechanism in the fintech industry's fight against fraud. However, the application of ML in fintech is not without challenges (*Kulatilleke, 2022*). Both the accessibility and quality of the data are important to the efficacy of ML models. It is essential to have access to credible and representative data sets that reflect all the aspects of counterfeit activity. Furthermore, ensuring data privacy and regulatory compliance is vital to protecting client information and preserving confidence in the fintech ecosystem. One of the significant challenges faced by the fintech industry, including the banking sector, is the constant threat of cyberattacks and data breaches. To protect customer data from such attacks, fintech institutions have adopted fraud detection tools and improved their existing fraud detection using ML algorithms (*Long et al., 2020*). Data breaches, in particular, pose a severe risk as they involve unauthorized access to sensitive personal information through computer systems, either by external attackers or insiders.

## Fintech with federated learning

Federated learning (FL) is one of the approaches to distributing machine learning that allows training a model on several decentralized devices/servers without leaving the data location. In particular, this approach is very useful where high privacy for data is needed, and the different data owners collaborating could train a global model without actually sharing the raw data. This section covers the general aspects of model aggregation techniques in FL.

Introduced by Google in 2017, FL ensures privacy and enhances the efficiency of data scientists' work (*Dash, Sharma & Ali, 2022*). By training models across multiple decentralized peripheral devices, such as Linux-based servers (Redhat, Ubuntu, *etc.*), without the exchange of raw data, FL provides significant benefits in terms of privacy, security, and avoiding the need to transmit sensitive data to a main server (*Li et al., 2020a*). Instead, the model is transmitted to the distributed devices, and the model is updated locally. FL enables ML engineers and data scientists to work effectively with decentralized data while prioritizing privacy. This not only safeguards the data privacy of banks and fintech institutions but also mitigates the risk of data security breaches. Privacy is a top concern in FL, as hostile attackers can impersonate model coordinators and utilize gradient-based privacy attack methods to infer user data, resulting in privacy leakage

(*Yang, Fan & Yu, 2020*). Within the fintech ecosystem, FL emerges as a privacy-preserving technology that expedites financial processes by allowing AI models to be trained collaboratively across decentralized devices, facilitating real-time insights and decision-making without requiring data to be transferred to a central database (*Dash, Sharma & Ali, 2022*). This approach ensures data privacy and minimizes the potential risks associated with data security breaches (*Yu et al., 2020*). The integration of FL framework strategies within the fintech industry provides numerous benefits. Its core purpose is to preserve user privacy while enhancing the efficiency of data scientists' work. By leveraging decentralized devices and servers housing local data sets, scientists can train models effectively and share statistical data analysis models. This approach grants scientists access to more robust models capable of capturing counterfeit activities.

### Model aggregation process

FL aggregation process generally involves the following steps:

1. Local training: Each client trains a local trained model on its private dataset.
2. Model update sharing: Clients send their current local model updates to a central server.
3. Model aggregation: Central server aggregates the updates to form a new global model.
4. Model distribution across the globe: As clients receive the updated global model, they use it for training purposes on local data.

Therefore, over a set of iterations, given a converging series, the global model could be improved progressively at each round of aggregation.

### Model aggregation techniques

Various techniques can be employed to aggregate the model updates from different clients:

1. **Federated averaging (FedAvg):** The simplest core aggregation method, where the central server averages model updates. This is normally directly proportional to the number of data points each client holds. FedAvg can be mathematically represented as:

$$w_{\text{global}} = \frac{1}{N} \sum_{i=1}^{N} n_i w_i$$

where $w_{\text{global}}$ is the global model, $n_i$ is the number of data points in the $i$-th client's dataset, $w_i$ is the model update from the $i$-th client, and $N$ is the total number of clients.

2. **Homomorphic encryption:** can be used at the core of secure aggregation techniques to improve privacy. In this way, these techniques ensure clients' privacy since the central server only receives the final result after aggregation and does not see the individual updates.

3. **Differential privacy:** This technique adds noise to the model updates in smart ways as they are about to be transferred to the central server, so the updates themselves do not transfer sensitive information about the clients' data.

4. **Hierarchical aggregation:** In large-scale federated networks, one can use hierarchical aggregation. For example, any intermediate server, such as an edge server, will aggregate

updates from a subset of clients before sending them to the central server. This reduces communication and thus postpones scalability.

Model aggregation is one of the basic building blocks of FL, describing how collaboratively trained AI models retain client data privately. Accordingly, FL efficiently puts together model updates from decentralized clients through federated averaging, secure aggregation, and differential privacy. However, communication efficiency, client heterogeneity, and fault tolerance might present severe problems in FL implementation if not carefully managed.

## Fintech with blockchain-based federated learning

Blockchain is a decentralized and distributed ledger technology that records transactions across a network of devices, ensuring transparency and immutability by creating a secure and immutable chain of data blocks. Blockchain-based FL is a novel technique that combines FL's benefits with the credibility and security of blockchain technology. This technique helps many individual applications and entities collaborate on ML models without exposing their data while keeping an immutable and valid record of all transactions and data transfers (*Nguyen et al., 2021*). These might help handle data privacy issues and regulatory limits, while also allowing financial institutions to use machine learning to improve their services (*Rizinski et al., 2022*). Blockchain-based FL enhances the accuracy and efficiency of financial models by allowing financial firms (*Lu et al., 2019*). Blockchain-based FL enhances the accuracy and efficiency of financial models by allowing financial firms to exchange data and information. By integrating data from several sources, financial institutions can construct more detailed and accurate models that can better detect fraud, forecast market movements, and improve risk management (*Li et al., 2020b*).

## MOTIVATION

Federated learning aims to train ML models across a number of remote devices or servers without the need to centralize or share real user data. Instead, only model updates (gradients) are shared. While this approach greatly protects user privacy, there are potential risks if not implemented correctly, such as the risk of privacy breaches. For instance, if model updates contain sensitive information or if malicious actors attempt to deduce user data from these updates (*Mammen, 2021*). To ensure privacy in FL, various methods can be employed, including safe aggregation, differential privacy, homomorphic encryption, and FL frameworks like *Tensorflow privacy (2021)*. These techniques are designed to safeguard model updates and prevent the reconstruction of individual user data from them (*Suvarna & Kowshalya, 2020*; *Yang et al., 2019*; *Singh et al., 2021*). This research aims to leverage ML techniques within a privacy-preserving framework for the detection of fraudulent credit card transactions. By considering privacy security measures, privacy risks through trust, and privacy governance mechanisms, we aim to enhance fraud detection while maintaining the privacy of sensitive client data. Utilizing client-server based blockchain FL allows us to achieve this goal effectively. ML has made significant

strides in the fintech industry, including substantial achievements in identifying fraudulent transactions using data-driven insights.

## CONTRIBUTION

In the overall contribution of this study, we present how smart contracts in blockchain-based FL play a vital role in ensuring privacy during the training of ML models across multiple decentralized peripheral devices, such as Linux-based servers (Redhat, Ubuntu, *etc*.), without requiring the exchange of raw data. The main contributions to this article are as follows:

1. Proposed Blockchain-based federated learning framework that offers continuous learning and improved fraud detection models while maintaining data privacy.
2. Implementation of smart contract for ensuring privacy during the training and sharing of machine learning models.
3. Implementing a global federated learning model to enhance fraud detection techniques by combining multiple FL models.

The rest of the article is organized as follows: "Literature Review" presents the literature review of the related works in FL and blockchain in fintech. "Methodology" discusses the methodology employed. "Proposed Framework" describes the proposed framework in detail. The use-case discussion is done in "Usecase Discussion". "ML Implementation on the Federated Network" gives an overview of ML implementation. "Discussion" presents the experimental results and related discussion. Finally, the article concludes in "Conclusions".

## LITERATURE REVIEW

In this section, we discuss an overview of the latest advancements in the field of fintech, specifically focusing on securing financial systems from counterfeit activities. We discuss the cutting-edge research and innovative approaches that have been undertaken to enhance the security measures in the fintech industry. Additionally, we delve into the implementation of FL, a privacy-preserving technique, to safeguard sensitive financial data while enabling collaborative model training. This aims to highlight the significant developments and progress made in securing fintech systems and protecting against counterfeit activities. The idea of FL was first presented by Google in 2017 as an essential drive pointed toward supporting data scientists in their work. The essential objective behind this idea was to give significant help to data scientists and engage them in their data-driven tries. FL arose as an original methodology that tended to the difficulties of data privacy and security by empowering cooperative model preparation while keeping delicate data decentralized and restricted. This innovative framework disrupted the domain by allowing data scientists to utilize combined knowledge and experiences from multiple data sources without compromising the privacy of individual data (*Dash, Sharma & Ali, 2022*). With the advent of new and emerging technologies, there is a growing need to ensure the security of customer data. One effective approach is the implementation of FL as a protective layer over various ML models. This enables the prediction of counterfeit

transactions more robustly and securely. *Long et al. (2020)* investigate how FL can be used in open banking to prevent fintech fraud while protecting data privacy and security. It highlights the collaborative nature of FL and its potential benefits in improving and boosting fraud detection without the need to disclose raw customer data. The authors' analysis in *Kagan (2020)* provides an outline of fintech's evolution and impact on the financial industry. It discusses the key components of fintech, as well as its benefits, drawbacks, and the necessity for rules to safeguard consumers and ensure system stability. The author also explored other major fintech principles and applications, including peer-to-peer financing, robo-advisors, blockchain technology, digital payments, and mobile banking apps. The impact that fintech has had on financial transactions—making them quicker, more effective, and available to a wider consumer base is also addressed. Several aspects of FL have been studied by *Ogundokun et al. (2022)*, including its application domains and blockchain integration. The study emphasizes ways FL protects privacy by allowing for cooperative model training without revealing raw data. It also covered how blockchain technology contributes to decentralized data protection for privacy. In their discussion of the value of data protection in the fintech sector, *Dash, Sharma & Ali (2022)* focus especially on personally identifiable information (PII). The advantages of FL in cooperative, model-based training, protecting data security and privacy, and drawing insightful conclusions from sensitive financial data are all covered and investigated. They highlight the difficulties, like the necessity for strong encryption methods and trust frameworks, as well as issues with communication security and efficiency. As a countermeasure against credit card fraud, *Yang et al. (2019)* provide FL for fraud detection (FFD). FFD enables banks to train their fraud detection models using locally distributed data from their databases. These locally computed model updates are aggregated to create a shared fraud detection system (FDS), allowing banks to benefit from a collective model without compromising the privacy of cardholders' information. The challenges of using blockchain technology for fraud detection in fintech were explored in *Bin Sulaiman, Schetinin & Sant (2022)*. The article discusses potential issues, including slowdowns, scalability problems, higher energy usage, operational inefficiencies, and costs. The complex nature of training data and privacy concerns in data collection are also acknowledged as challenges in the context of ML. *Varmedja et al. (2019)* focus on credit card fraud detection and the effectiveness of ML algorithms in classifying transactions as counterfeit or genuine. They utilize the Credit Card Fraud Detection dataset for their analysis, highlighting the application of ML in addressing a significant concern involving the loss or theft of credit cards or sensitive credit card information. *Rizinski et al. (2022)* highlight the ethical challenges posed by ML in fintech. They identify four key ethical concerns, including privacy, bias, transparency, and accountability in ML models. The article stresses the need to establish mechanisms to hold developers accountable for the decisions made by these models, given their significant impact on people's lives. *Barbu et al. (2021)* emphasize the significance of customer experience in fintech and its role in mitigating the risks of fraud. They delve into various factors, such as user interface design, ease of use, trust, transparency, and personalized services, as pivotal for building customer trust and preventing counterfeit activities. The importance of data security and privacy

measures is highlighted to safeguard customer information. The implementation of blockchain technology by fintech companies is regarded as the next crucial step in the industry's growth and sustainability. According to a recent mapping study (*Fernandez-Vazquez et al., 2019*), there is a deep focus on challenges such as security, scalability, legal and regulatory issues, privacy concerns, and latency in the adoption of blockchain technology within the fintech sector. However, proposed solutions for these challenges are still in the early stages of development and are far from being fully effective. The study also points out that the majority of research in this field is focused on the finance and banking sectors, with not much thought given to other industries that could play an important role in the continuing adoption of blockchain. *Nelaturu, Du & Le (2022)* also explored the applications of blockchain technology in the fintech field. It also provides a taxonomy for fintech ecosystems listing some implementation scenarios. Challenges related to blockchain integration in financial institutions are also listed.

The evolving landscape of fintech encompasses a wide range of applications, including online money transfers, crowdfunding, and investment management, underscoring the paramount importance of security and privacy measures, notably through Blockchain-based fintech applications. *Baliker et al. (2023)* systematically reviews recent advancements in Blockchain-based fintech applications while shedding light on the emerging cyber threats that have evolved alongside. *Raikwar et al. (2018)* present the design and performance analysis of a blockchain-enabled platform for automating insurance processes, utilizing smart contracts, and an experimental prototype on Hyperledger Fabric. Blockchain's integration into the insurance industry for enhanced transaction execution, payment settlement, and security is a transformative development. The rapid adoption of fifth-generation (5G) and Beyond 5G (B5G) networks has spurred an increase in edge computing, enabling extensive data collection and transmission from edge devices for big data analytics. This data fuels the advancement of artificial intelligence through high-quality ML models, with privacy concerns addressed through FL. To tackle these persistent challenges, the integration of blockchain-enabled FL and Wasserstein generative adversarial network-enabled differential privacy (DP) is proposed in *Wan et al. (2022)*, providing decentralized, secure, and efficient mechanisms for protecting model parameters in B5G networks. *Dai (2022)* introduces a blockchain-based decision-making system integrating FL and evolving convolutional neural networks, with applications in assemble-to-order services and the Metaverses. The research focuses on the development and evaluation of an optimal policy computation algorithm for smart contracts on the blockchain. *Kollu et al. (2023)* introduce a cloud-based intrusion detection system using IoT FL and smart contract analysis. The method employs a novel approach, requiring users to authenticate by creating a route on a world map. Evaluating with 120 participants, including 60 with fintech backgrounds, simulations in Python using various datasets showed promising results: 95% accuracy, 85% precision, 68% recall, and 83% F-measure, among others. The work proposed in *Yang et al. (2022)* introduces an efficient credit data storage mechanism paired with a deletable Bloom filter to ensure consensus during the training and computation process. Additionally, authority control and credit verification contracts are proposed for secure certification of credit sharing model results in FL. *Wan*

*et al. (2022)* suggests merging blockchain-enabled FL with WGAN-enabled differential privacy (DP) to safeguard the parameters of the model of edge devices in B5G networks. Blockchain facilitates decentralized FL, reducing communication costs between the cloud and edge, and mitigating data falsification issues. *Chatterjee, Das & Rawat (2023)* offers an FL-empowered Recommendation Model (FLRM) that leverages both FL and blockchain technologies. In FLRM, the central server manages model aggregation and communicates with the blockchain network. Financial organizations keep their data on private blockchains while participating in the FL process. *Noman et al. (2023)* discuss the challenge of developing accurate global classifying models in healthcare due to the lack and diversity of medical data. Privacy concerns and legal restrictions have hindered data sharing among healthcare institutions, making it imperative to develop methods that can learn from distributed, heterogeneous data. Leveraging FL and blockchain technology, the proposed mechanism ensures privacy while effectively training and aggregating local models. The model demonstrates that the performance of the federated model rivals that of single-source models, achieving a testing accuracy of 88.10% for five classes. *Xiao et al. (2023)* investigate how Blockchain and FL can be combined while complying with regulations like the General Data Protection Regulation (GDPR) and similar laws on data protection. They aim to provide a basis for creating applications that are both legally compliant and user-friendly across different fields that depend on user data. This integration seeks to improve data security and privacy within the boundaries of regulatory requirements.

A summary of related surveys is provided in Table 1 considering the existing solutions' architecture design taxonomy, which includes client-server architecture and peer-to-peer networks. Additionally, we focus on existing solutions related to privacy and security measures, such as those based on privacy security measures incorporating the CIA trait, privacy risk measurement through trust mechanisms, and privacy governance mechanisms. Some studies (*Mothukuri et al., 2021*; *Yin, Zhu & Hu, 2021*; *Zhang et al., 2021*; *Lyu, Yu & Yang, 2020*) examined traditional ML methods as well as FL privacy concerns, while others (*Jagarlamudi et al., 2023*; *Lyu et al., 2022*) examined certain privacy concerns about federated learning, including the implementation of privacy security measures and the calculation of privacy security risks to implement scenarios. *Beutel et al. (2020)* introduce Flower, a comprehensive decentralized FL framework that offers novel capabilities for conducting large-scale FL experiments and accommodates diverse FL device scenarios. Additionally, they focus on the implementation of privacy measures concerning data attributes shared by trained models. *Reina et al. (2021)* introduced the Open Federated Learning (OpenFL) framework. This framework facilitates the training of ML algorithms through the data-privacy-focused collaborative learning approach of FL. It emphasizes privacy and security measures within a centralized architecture tailored for FL environments. OpenFL is compatible with training pipelines that use both TensorFlow and PyTorch. Furthermore, it provides seamless extension capabilities for other ML and deep learning frameworks. The extensive literature review highlights the ever-evolving landscape of fintech and its profound impact on various industries, particularly in the context of security, privacy, and data protection. Integrating advanced technologies, such as FL and Blockchain, has emerged as a pivotal solution to address the inherent challenges

**Table 1 Comparison of existing work for architecture and privacy.**

| Existing survey | Year | Client-server architecture | P2P Architecture | Privacy protection | Privacy-risk assessment | Privacy governance |
|---|---|---|---|---|---|---|
| *Jagarlamudi et al. (2023)* | 2023 | | ✓ | ✓ | – | – |
| *Lyu et al. (2022)* | 2022 | ✓ | ✓ | ✓ | ✓ | |
| *Beutel et al. (2020)* | 2020 | | ✓ | ✓ | | ✓ |
| *Deng et al. (2022)* | 2021 | | ✓ | ✓ | | ✓ |
| *Reina et al. (2021)* | 2021 | ✓ | | ✓ | | ✓ |
| *Rafi et al. (2024)* | 2024 | ✓ | ✓ | ✓ | | |
| *Li et al. (2021)* | 2021 | ✓ | | ✓ | | ✓ |
| Our work | 2024 | ✓ | ✓ | ✓ | ✓ | ✓ |

of data privacy, security, and collaboration in fintech. These technologies enable collaborative model training while preserving individual data privacy, facilitating the detection of counterfeit activities, enhancing fraud detection, and ultimately improving customer experiences. However, they are not without their own set of challenges, including privacy, communication efficiency, and security concerns.

# METHODOLOGY

The research methodology is based on these segments, is built to support the next, and is crucial and interconnected with the others. The methodology of this research is divided into the following primary parts:

1. Dataset collection and exploratory data analysis.
2. Machine learning model implementation in fintech.

## Dataset collection & exploratory data analysis

To protect customer privacy and ensure the exclusion of personal or sensitive information, we have incorporated a credit card-related dataset obtained from Kaggle (*Kartik2112, 2020*). This dataset consists of credit card transactions labeled either as fraudulent or not fraudulent. It is an artificial dataset, generated using the Sparkov data generator, containing both numerical and categorical features. The Kaggle data used contains more than 1.8 million instances and over 20 attributes. After performing dimensional reduction, specific attribute selection was carried out to facilitate further analysis. Table 2 details the description of all variables together with the reason behind their inclusion in this dataset.

### Data collection

The first step in this methodology involves gathering the relevant data set for the study. Data sources are carefully identified, ensuring they are reliable and pertinent to the research question. The testing is carried out on the data set pulled from Kaggle. As shown in Fig. 2 the data set is derived from fintech sources, incorporating credit card details and other raw data typically found in the banking sector, with over 1.8+ million instances. The data set consists of two files, the training file consists of nearly 1.3+ million instances, and

**Table 2  Feature description and reason for inclusion.**

| Feature name | Type | Description | Reason for inclusion |
|---|---|---|---|
| TransactionID | Categorical | Unique identifier for each transaction | Identifies each transaction uniquely |
| TransactionDT | Numerical | Time from a reference datetime | Captures timing patterns of transactions |
| TransactionAmt | Numerical | Transaction amount | Identifies unusual transaction amounts |
| ProductCD | Categorical | Product code | Differentiates between types of transactions |
| card1–card6 | Categorical | Payment card information (type, category, bank, *etc.*) | Provides detailed card information |
| addr1, addr2 | Numerical | Address information | Helps detect location-based anomalies |
| dist1, dist2 | Numerical | Distance between transaction and cardholder's address | Identifies discrepancies in expected distances |
| P_emaildomain | Categorical | Purchaser email domain | Identifies unusual email domains for fraud detection |
| R_emaildomain | Categorical | Recipient email domain | Identifies unusual email domains for fraud detection |
| C1–C14 | Numerical | Count features | Indicates frequency of transactions |
| D1–D15 | Numerical | Time deltas between different interactions | Measures recency of transactions |
| M1–M9 | Categorical | Match features (*e.g.*, address match, card match) | Detects inconsistencies in transaction data |
| V1–V339 | Numerical | Vesta engineered rich features | Complex features capturing various transaction details |
| DeviceType | Categorical | Type of device used for transaction | Identifies device-related anomalies |
| DeviceInfo | Categorical | Information about the device | Identifies device-related anomalies |

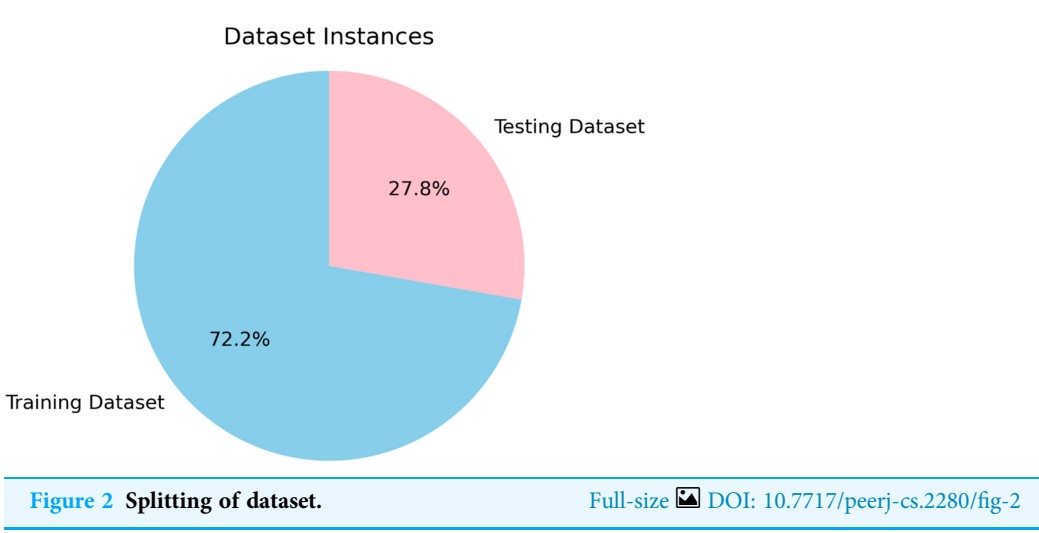

**Figure 2  Splitting of dataset.**               

the test file contains almost 0.5 million instances, which aggregate to make more than 1.8+ million instance data sets.

### Data pipeline

The data pipeline starts with the discovery, which identifies and understands the available data source. Once the data source has been identified, the next stage is data preparation. It involves cleaning, transforming, and integrating the data. Following the data preparation, the next pipeline moves into the model planning stage, where the objectives and goals of the analysis, selection of the appropriate ML algorithms, and design of the overall model architecture are done. The pipeline proceeds to the model-building phase, where the implementation of training of the selected ML algorithms using the prepared data. The

operation pipeline includes the deployment and integration of the models into real-world systems or applications. In the final stage, analyze, interpret, and communicate the generated insights and predictions to stakeholders.

### Tools and approach

Various tools are employed throughout the process to enable efficient data handling and analytics. These tools include Python for scripting, Pandas for data manipulation, Sklearn for different ML models Seaborn for graphical, and among others. We adopt a systematic and iterative approach to handle any unforeseen challenges in the data collection phase. For the FL implementation, the code is written on Java Spring Boot. Which typically consist of two modules, one is the federated server and the federated client(s). *Swagger* is implemented to look up the API endpoints.

### Data pre-processing

This step involves cleaning the data to eliminate noise, outliers, and inconsistencies. The data set undergoes standardization, normalization, missing data handling, and other necessary processes to ensure data quality and readiness for the subsequent phase. Firstly, categorical features such as transaction date, merchant, category, gender, city, state, *etc.*, are encoded into ordinal integers using the OrdinalEncoder. Here, it is preferable to use an OrdinalEncoder rather than one-hot encoding since it can reduce dimensionality, preserve interpretability, preserve ordinal relationships between categories, and use memory efficiently. Instead of creating several binary features for each unique category, which could result in problems like the curse of dimensionality, this approach gives each category a unique integer. Subsequently, numerical features undergo scaling using the MinMaxScaler, ensuring their values fall within a fixed range, between 0 and 1. This transformation ensures that all numerical features have the same scale, preventing features with larger magnitudes from dominating the model training process. Additionally, to address class imbalance issues in the dataset, the NearMiss under sampling (*Imbalance-Learn, 2023*) technique is applied. This technique reduces the number of majority class instances (potentially non-fraudulent transactions) to balance the class distribution with the minority class (likely fraudulent transactions). Only a subset of the samples from the majority class that are most similar to the minority class are kept by the algorithm. There are three variants of NearMiss: NearMiss-1, NearMiss-2, and NearMiss-3. While NearMiss-2 chooses samples that are farthest from the majority class, NearMiss-1 chooses samples from the majority class that are closest to the minority class. Here NearMiss-1 variant was utilized for the undersampling purpose.

### Exploratory analytics

Once the data is cleaned and pre-processed, it conducts exploratory data analysis (EDA) to understand the data set's underlying structure and relationships. It provides insights into data trends, correlations, and patterns that can inform feature selection and model selection. If the transaction is found to be counterfeit, it is marked as a fraud transaction. Figure 3 shows the distribution of the target variable (fraud) in the dataset before and after applying under-sampling. The incidence of fraud is observed to be significantly higher in the female

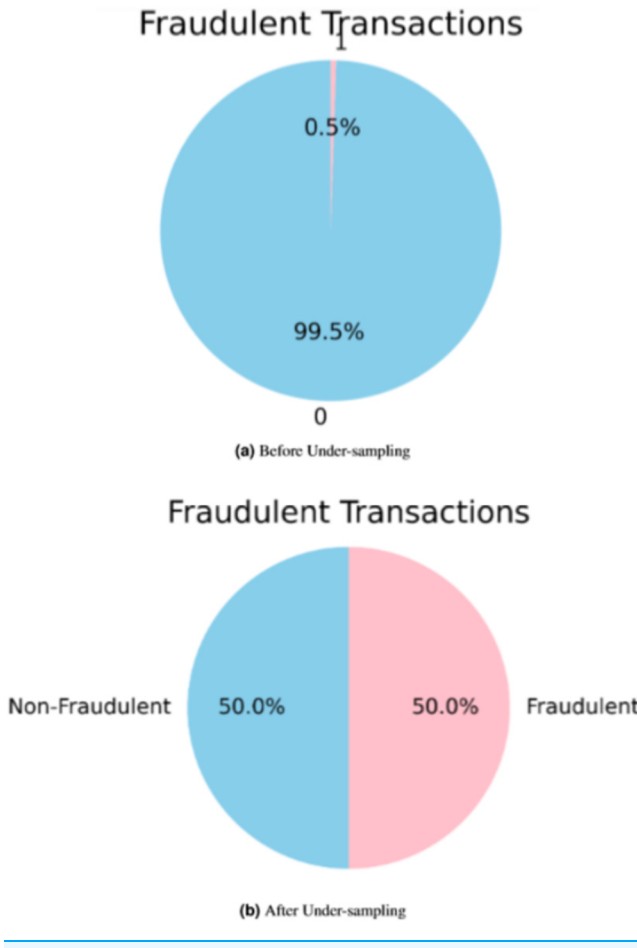

**Figure 3 Target (counterfeit) distribution.**

category compared to the male category, whereas the occurrence of fraud is relatively lower among males, as depicted in Fig. 4. The occurrence of credit card fraud across multiple job categories is shown in Fig. 5. The chart shows that certain job categories are more susceptible to credit card fraud than others. For example, workers in the retail and hospitality industries are more likely to be victims of credit card fraud than workers in other industries.

## Machine learning model implementation on fintech

To predict counterfeit transactions on credit cards, we have implemented six different ML models: the decision tree model (*Safavian & Landgrebe, 1991*), K-nearest neighbors (*Peterson, 2009*), support vector machine (SVM) (*Cortes & Vapnik, 1995*), Random Forest (*Breiman, 2001*), naive Bayes (*Rish, 2001*), and logistic regression (*Kleinbaum et al., 2002*) All these belong to supervised learning models. These models are applied to detect counterfeit transactions in fintech by training them on labeled data with features related to the transaction and the target variable indicating whether it is counterfeit or not. The implementation of these models is used to protect customers from counterfeit transactions and reduce the risk of counterfeit activities. To further improve the accuracy and performance of the models, these models are shared with the organization server, which

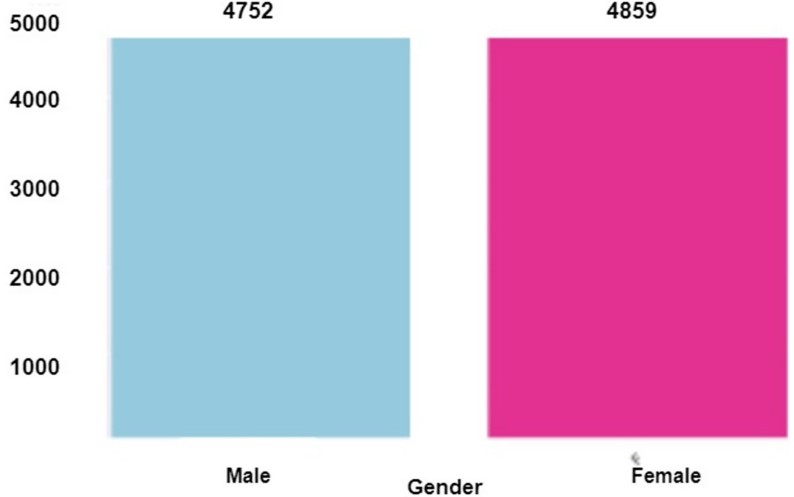

**Figure 4 Gender *vs.* fraud.**     

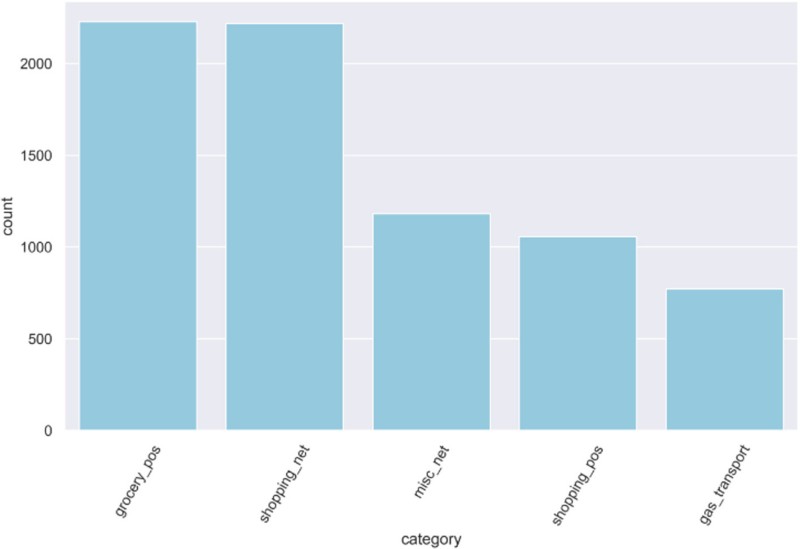

**Figure 5 Number of credit card frauds by job.**     

holds and versioned each model's parameters. These parametric values can be pulled by the client and implement the best parametric techniques. In this way, the accuracy can be enhanced without sharing the data set. which often results in better overall performance and more robust results. These ML models were applied to the dataset $D$. The modeling of which is as follows.

$$D = \{D_{train}, D_{test}\}_{(70,30)}$$
$$D_{train} = [transaction_1, transaction_2, transaction_3, \ldots, transaction_i]_{(i \to 1:1.3e6)} \qquad (1)$$
$$D_{test} = [transaction_1, transaction_2, transaction_3, \ldots, transaction_i]_{(i \to 1:5e5)}$$

### Decision tree model

The decision tree model (*Safavian & Landgrebe, 1991*) is a supervised ML algorithm widely used for tasks such as fraud detection, credit scoring, risk assessment, and financial analysis. It creates a flowchart-like structure where each branch represents a decision based on features, and each leaf node represents a prediction. The model recursively divides the data based on the best attribute or feature, guided by metrics like information gain or the Gini Index. This process continues until a stopping condition is met, ensuring adequate generalization. They are useful in fintech because they are interpretable and can analyze various sorts of data while capturing nonlinear relationships. Overfitting is possible, although it may be addressed with approaches like pruning and ensemble methods. The mathematical model for the Gini Index (*Karabiber, 2021*) when applied to our dataset $D$ is as follows:

$$\text{Gini}(D) = 1 - \sum_{k=1}^{n} (p_k)^2 \tag{2}$$

Here $D$ is a dataset that consists of samples from $K$ classes. The probability of a data sample belonging to class $k$ at a given node is denoted as $p_k$. whereas, $n$ denotes the number of data samples.

### K-nearest neighbors model

The KNN model (*Peterson, 2009*) is a widely used supervised machine learning algorithm for classification and regression. It determines the class label or value of a new data point based on its degree of similarity to the K-nearest neighbors in the training set. In classification, the majority of votes from the K-nearest neighbors determines the class label, but in regression, the projected value is the mean or weighted average of the target values. KNN is adaptive, does not require data distribution, and is implemented in Python using the Scikit-learn module. Cross-validation is used to determine the optimal number of neighbors $k$. The formula entails evaluating the distance between data points using criteria such as Euclidean distance or Manhattan distance, and then making predictions based on the $K$ closest neighbors. The Euclidean distance is calculated using the following formula:

$$d(\mathbf{D}_{\text{train}}, \mathbf{D}_{\text{test}}) = \sqrt{\sum_{i=1}^{n} (D_{\text{train}} - D_{\text{test}})^2}. \tag{3}$$

### Support vector machine model

The SVM model (*Cortes & Vapnik, 1995*) is an ML algorithm used for classification and regression tasks. It finds an optimal decision boundary that maximizes the margin between classes by separating data points. According to our problem, there are two classes $K = \{-1, +1\}$, for the dataset $D$ where $D$ belongs to the real numbers, and $d$ belongs to $D$. For $f(d) < 0$ signifies a class as $-1$ otherwise $+1$. We assume that both classes follow a

linear distribution. A function can be defined which can be used to distinguish between the two classes.

$$\text{Decision Function}: f(d) = (\mathbf{w^T d} + b) \tag{4}$$

where $\mathbf{w}$ is the weight vector representing coefficients for input features, and $b$ is the bias term providing an offset for the decision boundary.

SVM can handle linear and nonlinear relationships through the use of kernels. It is effective in high-dimensional feature spaces, less prone to over-fitting, and suitable for small to medium-sized data sets. However, SVMs can be computationally expensive and sensitive to hyperparameter choices. The decision function of SVM predicts the class or value based on the weighted sum of input features and a bias term. The objective of using the SVM model is effective in identifying complex fraud patterns because they can handle high-dimensional data. This is important because fraud data is often very complex and many different factors can contribute to fraud. By mapping the data to a higher-dimensional feature space, SVMs can identify patterns that would not be visible in the original data.

### Random forest model

The Random Forest model (*Breiman, 2001*) is a flexible ensemble approach that is useful for classification and regression application. To produce reliable predictions, it integrates several decision trees that have been trained on arbitrary subsets of the data. Random Forests reduce over-fitting, handle various types of features, and capture non-linear relationships. They provide accurate predictions through majority voting or averaging individual tree predictions. Random Forests also offer insights into feature importance. However, they can be computationally expensive and require careful parameter tuning. Random Forests are highly utilized in fintech due to their reliability and effectiveness in analyzing financial data. They are particularly useful for fraud detection as they can handle large and high-dimensional data sets while being resistant to over-fitting. Over-fitting, where a model is excessively tailored to the training data, is avoided by training multiple decision trees on different subsets of the data.

### Naive bayes model

The naive Bayes model (*Rish, 2001*) is a simple yet effective supervised ML algorithm. It implements Bayes' theorem under the presumption of feature independence, making it computationally efficient and well-suited to high-spatial data. Given a dataset $D$ and a class variable $K$, the naive Bayes model's mathematical model is as follows:

$$P(K|D) = \frac{P(D|K)P(K)}{P(D)}. \tag{5}$$

Within the fintech industry, naive Bayes models are useful for calculating probabilities and classifying data. They work with smaller training datasets, can accept both categorical and numerical variables, and are commonly used due to their simplicity and efficiency.

However, their feature independence assumption may limit their efficacy in cases involving correlated attributes.

### Logistic regression

Logistic regression (*Kleinbaum et al., 2002*) is a popular classification technique, that examines the relationship between independent factors and a binary outcome variable. It helps in examining the possibility of an event occurring, allowing financial institutions to develop models that detect abnormal trends and predict the probability of fraud. Due to its flexibility and compatibility with a variety of data sources and systems, it serves as an integral tool for enhancing fraud detection and prevention within the fintech sector. The attributes of the transaction, such as the transferred amount, time of occurrence, and location, are considered independent variables. The transaction's label, which indicates whether or not it is fraudulent is a dependent variable.

## PROPOSED FRAMEWORK

We have utilized blockchain technology to establish a client-server model within the proposed framework, as illustrated in Fig. 6. In this setup, the server node assumes the role of a central entity responsible for overseeing and controlling diverse learning models. Meanwhile, the client node functions as a host for numerous financial applications. The server's task includes providing a way to integrate many trained models without requiring data sharing. In the following section, we will delve into the technical aspects of the sub-modules included in the proposed framework.

### Communication between server and client nodes

The decentralized federated server and clients communicate through APIs to synchronize and update their models while protecting customer data. This approach ensures that only the models are shared, not the raw data sets, providing security for client nodes such as banks and preventing potential cyber-attacks aimed at accessing sensitive customer data. Decentralized FL effectively addresses the challenge of data leakage by training and operating ML models on separate and private data sets owned by individual clients. In this framework, we specifically used six different learning models: KNN, SVM, decision trees, naive Bayes, logistic regression, and Random Forest. This approach ensures that no sensitive information is shared within the proposed framework. Only the trained models are shared with the organization's central entity, the federated server. The federated server collects and aggregates the trained model information from all the clients, enabling the clients to request and incorporate the updated models as needed. This collaborative and secure framework facilitates efficient model-sharing and improvement without compromising customer privacy. In the following subsection, we will discuss the smart contracts and their execution processes residing in server and client nodes.

### Blockchain-based federated server

The blockchain-based federated server is responsible for managing and controlling global model updates throughout the learning process. To maintain a consistent and up-to-date representation of the trained models from blockchain-based federated clients, the

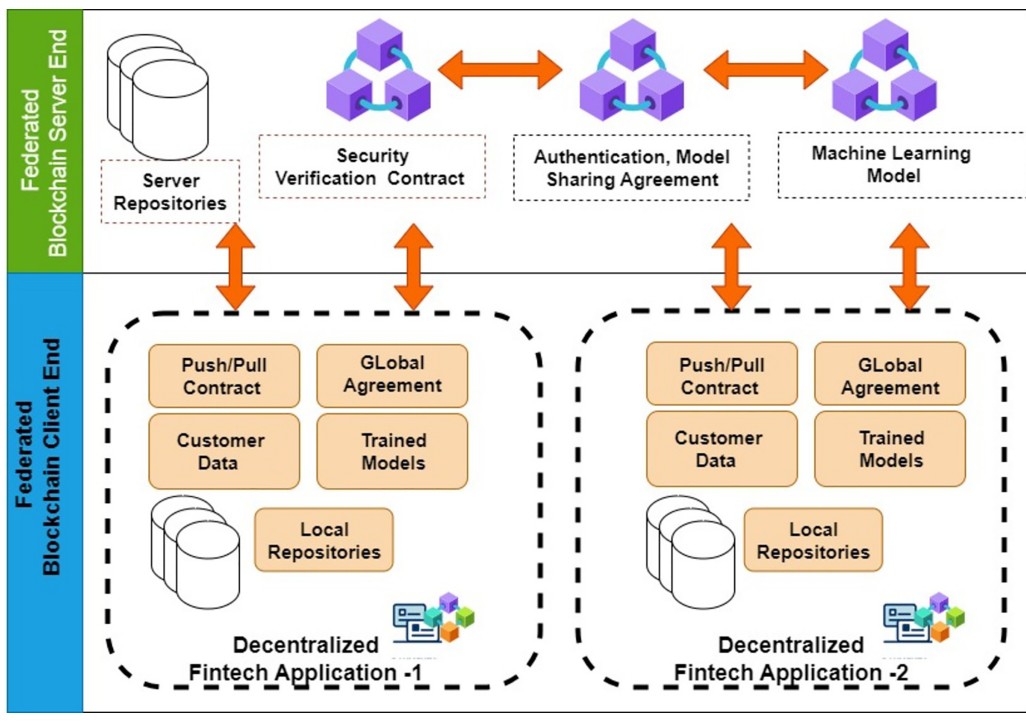

**Figure 6 Proposed framework: architecture design.**

---

**Algorithm 1 Security validation contract.**

**Input:** $\alpha_i$ and, $\gamma_i$

Where $\alpha_i \leftarrow Public_{Keys}$ and $\gamma_i \leftarrow Federated_{Challenge}$

**Output:** Legitimate or Not legitimate

**Step 1:** Verify the legitimacy of client node

**Step 2:** Send puzzle to the federated client

**Step 3:** Receive the puzzle output

**Step 3.1:** Compute SHA256*(Puzzle)*

**Step 3.2:** $\beta =$ SHA256*(Puzzle)*

**Step 4:** Validation of $\beta$ from blockchain

**Step 4.1:** Validation is successful access grant

---

federated server is responsible for retrieving and synchronizing trained model data from the client nodes. The server also oversees the verification of the legitimacy of client nodes through security validation smart contracts while ensuring data privacy when the client pushes its trained model to the server nodes.

### Security validation

Algorithm 1 represents the implementation of the 'Security Validation Contract' in which $\alpha_i$ represents the public key of the client federated node. The public keys of federated server

| **Algorithm 2** **Authentication and sharing agreement contract.** |
|---|
| **Input:** Receive request |
| **Output:** Generate the policies and send it to the client nodes |
| **Step 1:** Receive request from client nodes |
| **Step 2:** Create JSON policy file (JSP) |
| **Step 3:** Generate transaction and store in the blockchain |
| **Step 4:** Send the transaction hash to the client nodes |

and client nodes are fetched from a secure Lightweight Directory Access Protocol (LDAP) (*Howes, Smith & Good, 2003*) for authentication purposes. $\gamma_i$ represents the challenge created by the federated server when a client node wishes to authenticate with the server node. The client nodes first solve the puzzle and send the solution back to the server node. The server node verifies the received solution of the sent puzzle to grant access to the server nodes in the form of a registration token. Furthermore, the federated server holds the active update versions of the models obtained from multiple federated clients. It acts as a repository for these models, allowing seamless integration and exchange of model updates among the clients. The federated server comprises these main smart contracts; below is a discussion of these smart contracts and their workflow.

In the context of a federated server workflow, after successful authentication and access to federated server nodes, client nodes request authentication and model-sharing agreements between clients, as well as between clients and the server. This agreement is responsible for managing the authentication and data-sharing policies between client-to-client and client-to-server nodes.

### Authentication and sharing agreement

Algorithm 2 represents the 'Authentication and Sharing Agreement Contract' that is responsible for establishing authentication and model-sharing agreements between federated servers and client nodes. In this agreement, all entities define their security authentication mechanisms for their clients and their learning model-sharing attributes. Authentication plays a crucial role in identifying and verifying the identity of federated client node users to ensure that requests originate from authorized entities rather than unauthorized or malicious sources. Federated client nodes generate requests to federated server nodes to create these authentication and sharing policies after successful validation of both client and server federated nodes at the time of registration. Client nodes send the request messages to the federated server node using the POST method *via* Hypertext Transfer Protocol Secure (HTTPS) (*Cremers et al., 2017*). HTTPS is a secure iteration of HTTP that employs Secure Sockets Layer (SSL) (*Oppliger, 2023*) or Transport Layer Security (TLS) (*Krawczyk, Paterson & Wee, 2013*) to encrypt data transmitted between a web browser and a web server. Figure 7 illustrates the workflow of the authentication and sharing agreement contract. The server nodes create the policies and store them on the federated server node's blockchain for validation purposes. They also send the contract to

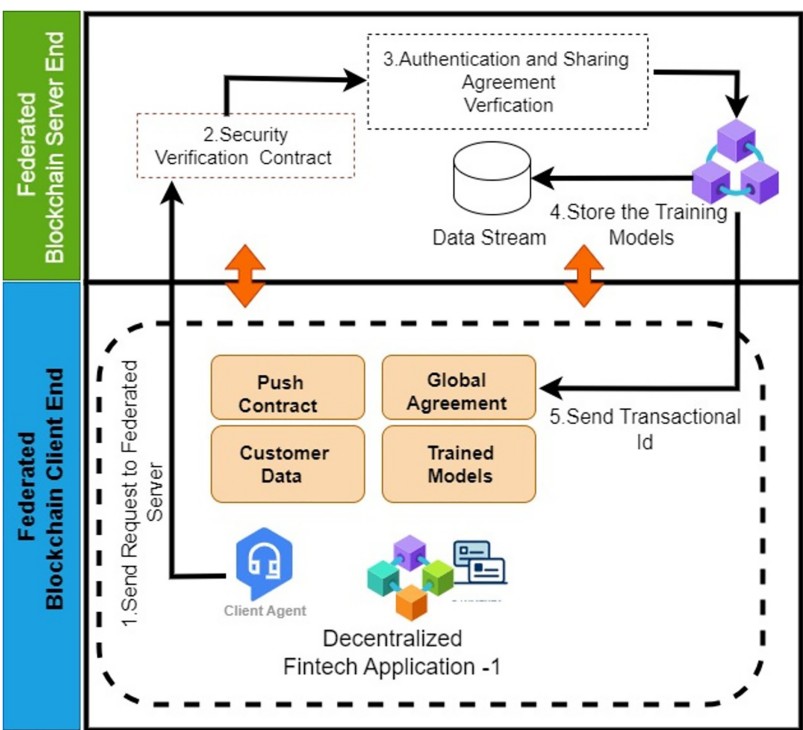

**Figure 7 Workflow diagram: authentication and sharing agreement.**

the client nodes in the form of a JSON web token (JWT) based token with an expiration date. This token must be included in the header of every call made by participants within a specified timeout period of 24 h. The timeout can be adjusted according to the policies made by the governing body.

### Add federated client training model

Algorithm 3 represents the 'Add Federated Client Training Model Contract' that is responsible for maintaining the training models of client nodes. The model is pushed from the client node to the server node. Initially, the server nodes verify the legitimacy of both client and server nodes through a security contract. After successful validation, the federated server nodes validate the authentication and sharing attributes of the federated server nodes. Upon successful validation, the hyper-parameters of the training model are stored in the blockchain by generating the transaction command. The federated server forwards the transaction ID (TID) to the client node for validation at the client's end. Figure 8 illustrates the workflow of the smart contract for adding client nodes' training models to the federated server nodes. Client nodes send the training model parameters in the form of a request message. The server nodes first verify the legitimacy of the client nodes' connection with the help of the security validation contract. Then, they proceed to the next steps, where federated server nodes verify the policies. After successful verification of authentication and security policies, the federated server stores the hyper-parameters of the training models in the blockchain using a transaction command. This algorithm

| Algorithm 3 | Add federated client training model contract. |
|---|---|

**Input:** Receive training model parameters

**Output:** Store the training model parameters

**Step 1:** Receive training model parameters in json

**Step 2:** Parse the message

**Step 3:** Search the repository of particular model

**Step 4: If** Search model is found

**Step 4.1:** Call the synchronization contract

**Step 4.2: Else** Create the new profile

**Step 5:** Store the models in the blockchain

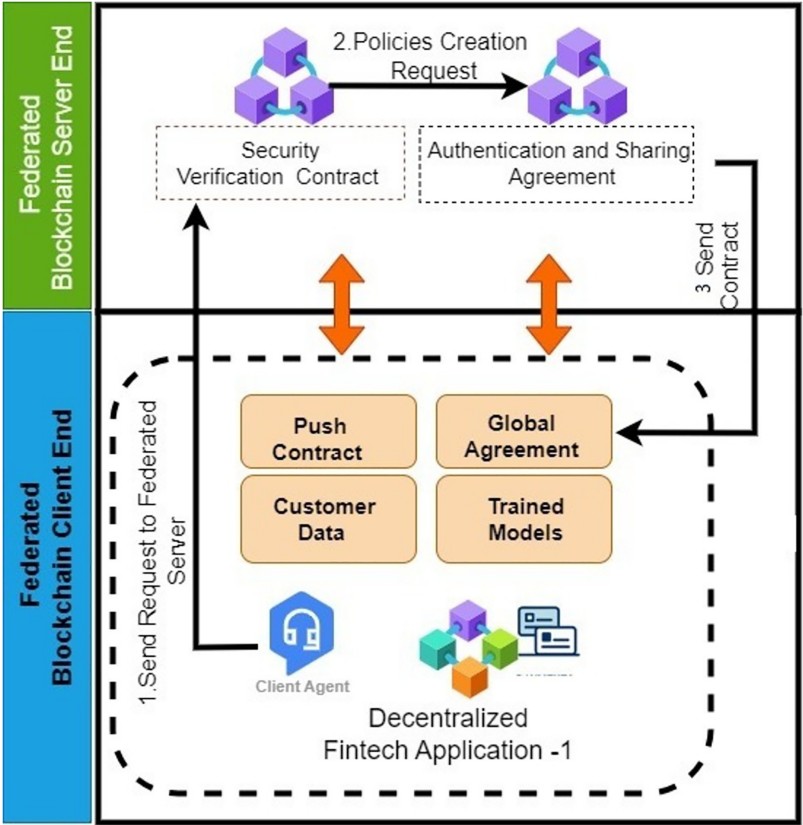

**Figure 8 Sequence diagram: federated server federated client training model.**

demonstrates the implementation of adding training models from multiple clients to the decentralized federated server nodes. The process begins with the server nodes parsing the incoming message and searching for the requested model in the repository. If the model is found, the server then triggers the synchronization smart contract. The synchronization smart contract is responsible for obtaining the updated training model from the client

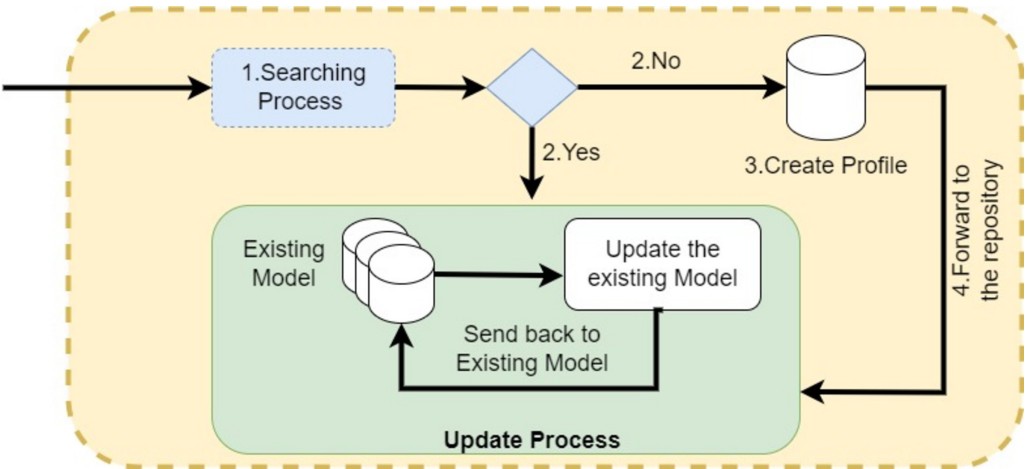

**Figure 9  Workflow of synchronization smart contract.**

node, which is continuously refreshed with new data. This ensures that the federated server always uses the latest model version, allowing it to keep track of model changes and manage different versions effectively. In the event that the training model is not found in the repository, a new model profile is created. Once the insertion is complete, the federated server notifies the client nodes of its success by returning the TID. This straightforward procedure enables the seamless addition of numerous ML models, making the system highly flexible for the incorporation of additional models as needed. Figure 9 shows the workflow of synchronization smart contract.

## Blockchain-based federated clients

Decentralized federated client nodes act as bank entities responsible for training ML models tailored to their specific environments to preemptively detect fraudulent activities. The federated server, instead of sharing raw user data, retrieves the updated models from these decentralized federated bank clients and stores them. Each client node hosts its local dataset, which includes private customer information and uses six different ML approaches like KNN, SVM, decision trees, naive Bayes, logistic regression, and Random Forest. in their respective local repositories. The Federated client nodes are comprised of these main smart contracts below is a discussion of their smart contracts with their workflow. In the context of a federated client workflow, the FL framework ensures that bank clients do not directly share their data but instead share their trained models with the federated server. This model-centric approach safeguards customer privacy. To preserve data privacy, the federated clients locally train their ML models on their respective private customer data sets. They periodically pull the updated model versions from the federated server, incorporating the collective knowledge of all clients' models. This collaborative learning process allows each client to benefit from the aggregated insights while still maintaining the confidentiality of their own data. After training the models locally, the federated clients send their model updates, rather than the raw data, to the federated server for aggregation. The federated server combines these updates and applies mechanisms to refine a global model. The federated server ensures that the updated model does not

| Algorithm 4 Add local trained model contract. |
| --- |
| **Input:** Customer data |
| **Output:** Store the trained models |
| **Step 1:** Validate the digital signature on customer data |
| **Step 2: If** Successful validation |
| **Step 2.1:** Convert the raw data in to CSV file |
| **Step 2.2:** Perform data pre-processing technique |
| **Step 2.3:** Trained data available learning model |
| **Step 3:** Store the models in the blockchain |
| **Step 4: Else** Return |

contain any sensitive customer information. This iterative process of model updates continues, enabling the federated clients to collectively improve the global model's performance without compromising the privacy of individual customer data. In the following subsection, we will discuss the workflows of the smart contract that resides in the client nodes.

### Add local trained model

Algorithm 4 represents the 'Add Local Trained Model Contract' that is responsible for the federated client nodes maintaining a repository of customer data, with each dataset registered in the local blockchain. These registrations are accompanied by the digital signature of the local federated client, which is stored in the Federated client blockchain for validation purposes. After validating the digital signature from the local blockchain, the customer dataset undergoes data pre-processing and is then trained using any of the registered learning models. After training, the processed data is stored in the repository.

### Smart push and pull

Algorithm 5 represents the 'Smart Push and Pull Contract' that is responsible for forwarding the latest tuned parameters of the trained model to the federated server and pulling the requested aggregated trained model to the decentralized federated server for use by the federated client nodes. Security validation is done in the same way as in previous sections.

## Security layers to ensuring privacy, trust, and governance

The proposed blockchain-based FL system involves several security layers to ensure privacy and maintain trust

1. **Encryption:** Static cryptographic techniques encrypt all data transmitted between nodes in the FL network. This encryption ensures data remains protected and private during transmission and storage.
2. **Privacy security measure:** Privacy techniques augment individual data samples with noise before transmitting them to the central server for aggregation. This prevents the

| Algorithm 5 | Smart push and pull contract. |
|---|---|

**Input:** Tuned parameters with aggregated trained model

**Output:** Updated global model, Acknowledgment

**Step 1:** Initiate 'Push' operation

**Step 1.1:** Forward tuned parameters to federated server node

**Step 1.2:** Include essential security parameters

**Step 1.3:** Verify the digital signature

**Step 1.4: If** Validation successful

**Step 1.4.1:** Update global model with received parameters

**Step 1.4.2:** Send acknowledgment of successful update

**Step 1.5: Else** Terminate operation and notify failure

**Step 2:** Initiate 'Pull' operation

**Step 2.1:** Client request on-demand global parameters

**Step 2.2:** Include security parameters in the request

**Step 2.3:** Validate the digital signature on the request

**Step 2.4: If** Validation successful

**Step 2.4.1:** Send the requested aggregated trained model

**Step 2.5: Else** Terminate operation and notify failure

extraction of non-essential information for model training while preserving the privacy of individual data points.

3. **Privacy security risk:** Trust-based secure aggregation protocols accomplish the aggregation of model updates from multiple participants while protecting their privacy. These protocols ensure that the central server can compute aggregate statistics while preserving the anonymity of each participant's contribution.

4. **Governance mechanism:** A governance mechanism is established through client and server architecture to oversee the operation of the FL network and enforce privacy and security policies.

## USECASE DISCUSSION

To test the feasibility of the proposed framework, we consider a use case in which Federated ClientNode-1 develops an ML model for fraud detection using a dataset of their private customers. ClientNode-1 predicts the correct counterfeit transaction. ClientNode-1 updates the federated server regarding the counterfeit attack, and ClientNode-1 updates the optimized hyper-parameters in the local blockchain. A Federated server complete with the client's digital signature for validation and storing the hyper-parameters. The Federated Server requests all the connected ClientNode to pull the latest optimized hyper-parameters so that all the other ClientNode are able to predict the same pattern of counterfeit if occurs at their end. ClientNode can pull the updated hyper-parameters in a

**Table 3  Model parameter configurations.**

| KNN model | | Decision tree model | |
|---|---|---|---|
| v1 | n_neighbors = 5, metric = 'minkowski', $p = 2$ | v1 | max_depth = None, random_state = None |
| v2 | n_neighbors = 5, metric = 'minkowski', $p = 1$ | v2 | max_depth = 25, random_state = 50 |
| v3 | n_neighbors = 5, metric = 'minkowski', $p = 2$ | v3 | max_depth = 50, random_state = 100 |
| v4 | n_neighbors = 10, metric= 'minkowski', $p = 1$ | v4 | max_depth = 75, random_state = 42 |
| v5 | n_neighbors = 10, metric = 'minkowski', $p = 2$ | v5 | max_depth = 100, random_state = 42 |
| **Naive bayes model** | | **SVM model** | |
| v1 | var_smoothing = 1e-8 | v1 | shrinking = True, random_state = None |
| v2 | var_smoothing = 1e-8 | v2 | shrinking = False, random_state = 50 |
| v3 | var_smoothing = 1e-7 | v3 | shrinking = False, random_state = 100 |
| v4 | var_smoothing = 1e-5 | v4 | shrinking = False, random_state = 42 |
| v5 | var_smoothing = 1e-3 | v5 | shrinking = True, random_state = 42 |
| **Logistic regression model** | | **Random forest model** | |
| v1 | fit_intercept = True, random_state = None | v1 | n_estimators = 100, random_state = None |
| v2 | fit_intercept = False, random_state = 50 | v2 | n_estimators = 10, random_state = 50 |
| v3 | fit_intercept = False, random_state = 100 | v3 | n_estimators = 20, random_state = 100 |
| v4 | fit_intercept = False, random_state = 42 | v4 | n_estimators = 25, random_state = 42 |
| v5 | fit_intercept = False, random_state = 42 | v5 | n_estimators = 30, random_state = 42 |

secure way. In this way, all the ClientNode(s) are now better equipped to apply the fraud detection model.

# ML IMPLEMENTATION ON THE FEDERATED NETWORK

In this work, we present a Federal learning-based system that preserves data integrity while allowing various organizations to train ML models on their own datasets. Each participant (bank) in the suggested framework is required to train a local model using its own dataset. After that, a global parametric model that gets employed for forecasting is created by combining the local models. This is carried out in a manner that ensures data security and confidentiality. Every bank or business uses a different ML model, and every model has a different set of parameters. The parameters were tuned to optimize the performance of the model on the specific dataset that the bank or entity had. Let's assume, that one bank might have used a decision tree model with 100 max depth, while another bank might have used the same model with 20 max depth and 10 features randomly sampled at each split. The specific parameters that were used would depend on the specific dataset that the bank or entity had, and the specific goals that the bank or entity was trying to achieve. The different parametric tuned models can be aggregated at the federated server node. This means the models can be shared and combined to create a more powerful model. Banks can then use this model to detect counterfeit activities. We have implemented six machine-learning models on five communication iterations between the federated server and clients. These five iterations are referred to as the version. From this point forward, the term "version" will refer to the iteration between the federated server and clients. Table 3 provides a

**Table 4 Train & test accuracy result.**

| Model | Version | Train accuracy | Test accuracy |
| --- | --- | --- | --- |
| Decision tree | v4 | 100% | 95.72% |
| Random forest | v1 | 99.99% | 95.07% |
| Naive bayes | v1 | 84.67% | 83.78% |
| Logistic regression | v2 | 79.86% | 79.24% |
| SVM | v5 | 77.67% | 74.33% |
| KNN | v2 | 80.76% | 69.46% |

**Table 5 Precision, recall, and F1-score.**

| Models | Version | F1-Score | Precision | Recall |
| --- | --- | --- | --- | --- |
| Decision tree | v4 | 1 | 0.9603 | 0.9537 |
| Random forest | v1 | 0.999 | 0.9624 | 0.9375 |
| Naïve bayes | v1 | 0.8274 | 0.9354 | 0.7258 |
| Logistic regression | v2 | 0.7896 | 0.825 | 0.7424 |
| SVM | v5 | 0.7617 | 0.7806 | 0.6771 |
| KNN | v2 | 0.788 | 0.7458 | 0.5908 |

comprehensive overview of the implemented ML models, including KNN, decision trees, naive Bayes, SVM, logistic regression, and Random Forest. Each model is presented in multiple versions (from v1 to v5) with corresponding parameter configurations. These parameter changes are necessary to understand and compare the behavior of each model in different scenarios.

## RESULTS AND DISCUSSION

The results are summarised in Table 4. With an accuracy of 95.72%, the decision tree model demonstrated the best performance. The Random Forest model was second, with an accuracy of 95.07%. The naive Bayes and logistic regression models performed moderately, with accuracies of 83.78% and 79.24%, respectively. The SVM and KNN models performed the least well, with accuracies of 74.33% and 69.46%, respectively. Additional statistical analysis was conducted to calculate the precision, recall, and F1-score of the applied ML models. The results are summarized in Table 5. Once again, the decision tree model outperforms the others, exhibiting the highest scores across all three parameters, with the Random Forest model closely following behind. In contrast, KNN and SVM have the poorest statistics in terms of precision, recall, and F1-score showcasing they are not efficient in the classification.

The version mentioned in the above result table has been trained with different parameters to get the best possible result. The parametric values of each model are tuned to maximize the model's ability to correctly predict the outcome of the data. The best versions of the ML models were combined at the federated server as shown in Table 6.

**Table 6 Parametric values version.**

| Model | Version | Parametric values |
|---|---|---|
| Decision tree | v4 | max depth = 75, random state = 42 |
| Random forest | v1 | n estimators = 100, random state = None |
| Naive bayes | v1 | priors = None, var smoothing = 1e-09 |
| Logistic regression | v2 | fit intercept = False, random state = 50 |
| SVM | v5 | shrinking = True, random state = 42 |
| KNN | v2 | n neighbors = 5, metric = 'minkowski', $p$ = 1 |

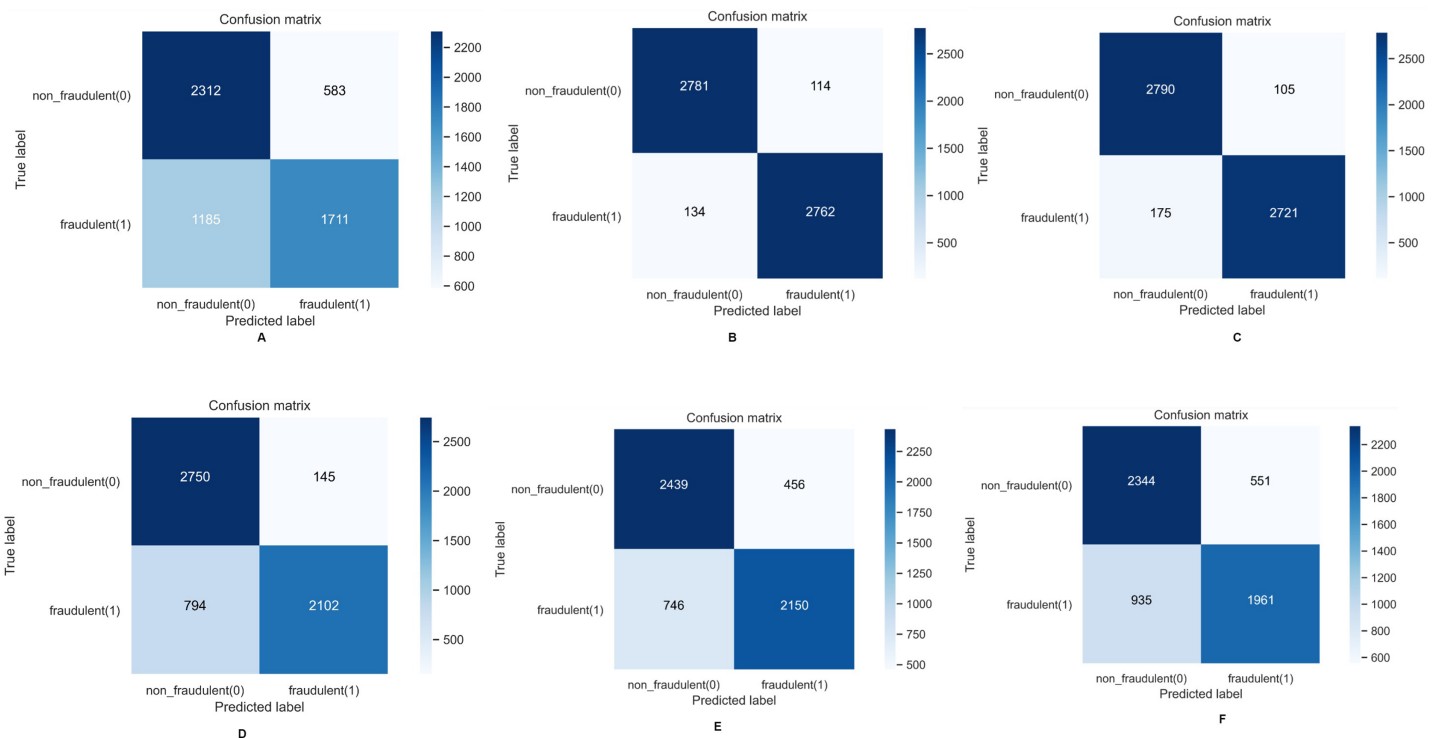

**Figure 10 Confusion matrices of machine learning models.**               

These results suggest that both the decision tree model and the Random Forest model are effective at detecting counterfeit transactions. However, the decision tree model appears to be slightly more effective than the Random Forest model.

The confusion matrices of the comprehensively implemented ML models using the FL implementation are shown in Fig. 10. It further confirms our previous observation that decision tree and Random Forest are the two best-performing ML models in our proposed approach. We have built and tested five different versions of the model. In each version, banks or entities participate in tuning the model parameters on their private premises. The federated server then collects all the models from the banks or entities and combines them at a central node. This allows any bank or entity that wants to update the ML models to
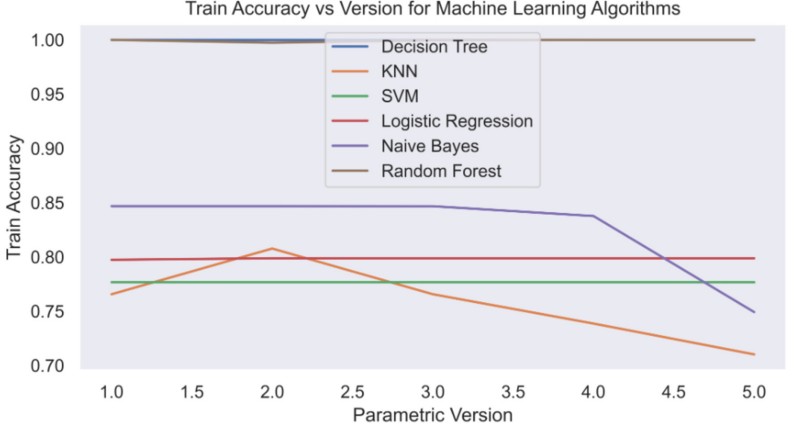

**Figure 11 Train accuracy *vs*. version.**

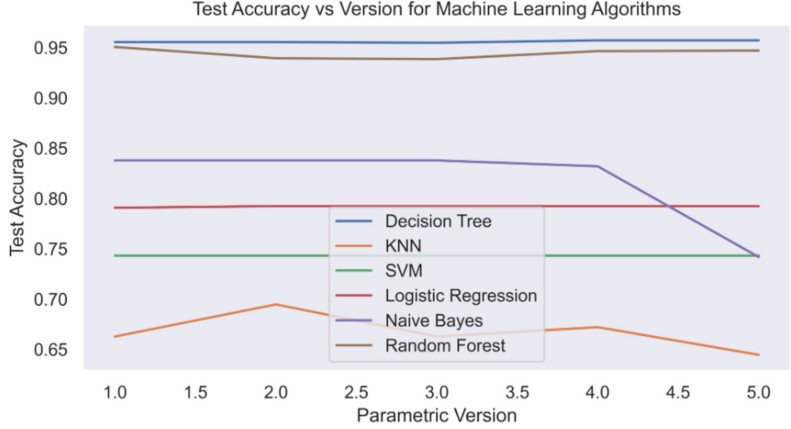

**Figure 12 Test accuracy *vs*. version.**

easily and securely pull them from the federated server node. The parameters of the models are tuned to give the best possible outcomes. Figure 11 illustrates the training accuracy of each version of the ML model. The training accuracy reflects the percentage of instances correctly detected during the training phase, which provides information about the model's performance on the training dataset. Examining training accuracy reveals how well the models learned from the training data and how well they fit the training set. We can assess the capacity to extract patterns from training data by comparing training accuracy across model versions.

Figure 12 illustrates the testing accuracy for each model version. The test accuracy is critical in fraud detection since it shows the model's capacity to generalise to new data and detect previously unknown counterfeit transactions. It serves as a model comparison benchmark, allowing fintech companies to make more informed judgements, optimise fraud detection tactics, and improve fraud protection measures.

A statistical computational analysis was performed to further evaluate the model's efficiency. The choice of hyperparameters significantly influences both the execution and

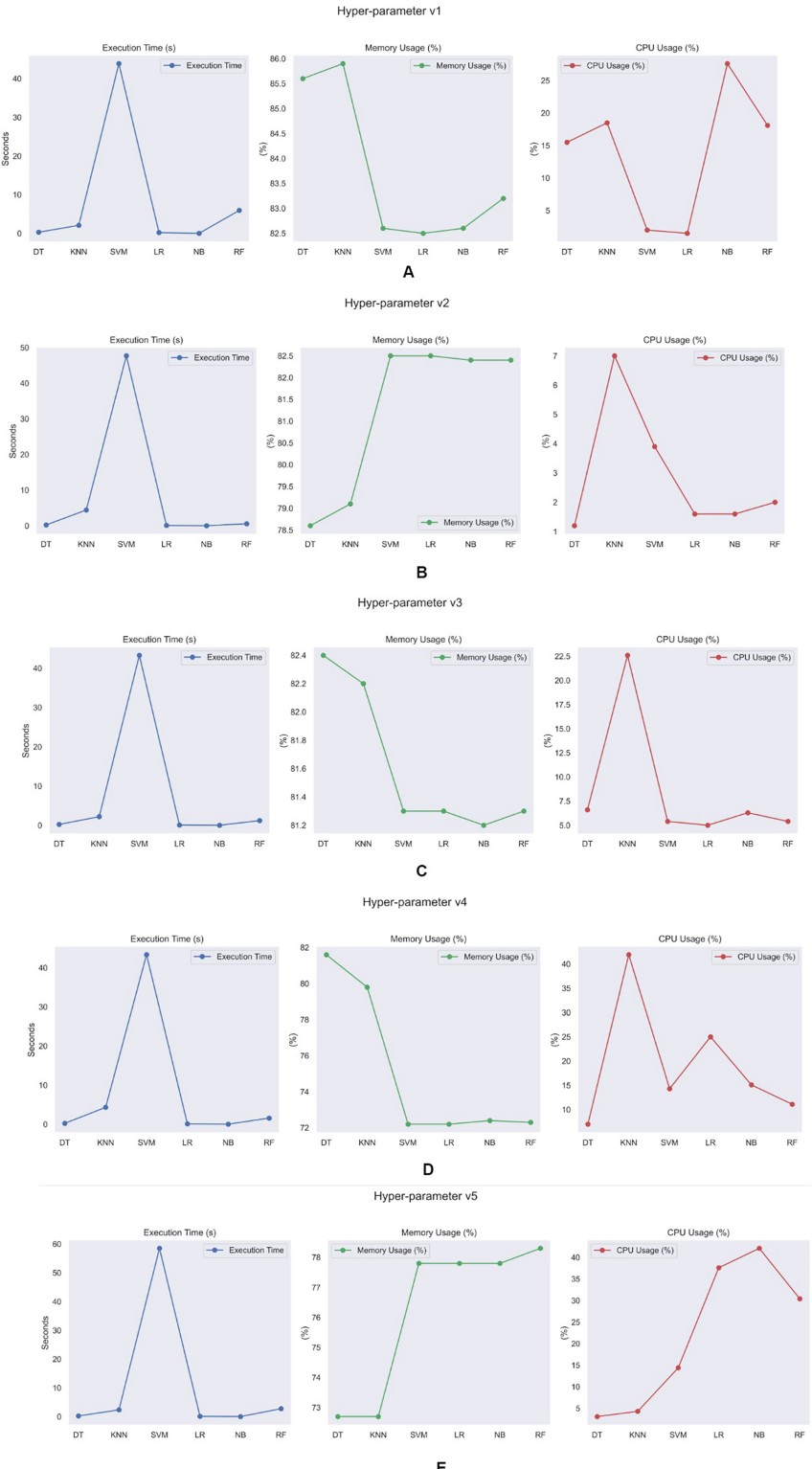

**Figure 13 Models computational analysis for each hyperparameter version.**

**Table 7 Performance analysis hyperparameter v1.**

**Hyperparameter version 1**

|  | Memory usage (%) | CPU usage (%) | Execution time (secs) |
|---|---|---|---|
| DT | 83.9 | 18.1 | 0.29 |
| KNN | 83.9 | 10.5 | 2.25 |
| SVM | 83.5 | 12.1 | 47.70 |
| LR | 83.4 | 16 | 0.21 |
| NB | 83.5 | 12.5 | 0.03 |
| RF | 84.2 | 11.5 | 6.17 |

**Table 8 Performance analysis hyperparameter v2.**

**Hyperparameter version 2**

|  | Memory usage (%) | CPU usage (%) | Execution time (secs) |
|---|---|---|---|
| DT | 79.5 | 12.9 | 0.26 |
| KNN | 79.4 | 10.3 | 4.30 |
| SVM | 78.9 | 7.7 | 47.28 |
| LR | 78.8 | 14.1 | 0.09 |
| NB | 78.8 | 18.2 | 0.03 |
| RF | 78.1 | 45.4 | 0.88 |

**Table 9 Performance analysis hyperparameter v3.**

**Hyperparameter version 3**

|  | Memory usage (%) | CPU usage (%) | Execution time (secs) |
|---|---|---|---|
| DT | 74.3 | 18.1 | 14.60 |
| KNN | 74.2 | 10.5 | 12.10 |
| SVM | 78 | 12.1 | 4.70 |
| LR | 78.1 | 16 | 3.80 |
| NB | 77.9 | 12.5 | 11.90 |
| RF | 77.8 | 11.5 | 21.20 |

efficacy of the training process. Accurate tuning and optimisation of hyperparameters are crucial for maximising model performance, reducing training time, and optimizing resource utilisation. Additionally, it is essential to understand the dynamic relationship between hyperparameters and the training process to construct effective ML models.

Figure 13 shows the performance indicators of training model execution time, CPU utilization, and memory performance across server nodes with hyperparameters of training models with five different hyperparameter tuning as shown in Tables 7–11. Decision trees, logistic regression, and naive Bayes have significantly faster execution times

**Table 10 Performance analysis hyperparameter v4.**

**Hyperparameter version 4**

|  | Memory usage (%) | CPU usage (%) | Execution time (secs) |
| --- | --- | --- | --- |
| DT | 77.9 | 46.1 | 0.28 |
| KNN | 78.6 | 5.8 | 4.96 |
| SVM | 75.5 | 4.2 | 46.09 |
| LR | 75.5 | 3.1 | 0.08 |
| NB | 75.4 | 5 | 0.02 |
| RF | 75.7 | 3.1 | 1.45 |

**Table 11 Performance analysis hyperparameter v5.**

**Hyperparameter version 5**

|  | Memory usage (%) | CPU usage (%) | Execution time (secs) |
| --- | --- | --- | --- |
| DT | 76.3 | 4.3 | 0.24 |
| KNN | 76.5 | 6.2 | 2.18 |
| SVM | 81.9 | 28.7 | 76.86 |
| LR | 81 | 33.3 | 0.09 |
| NB | 81 | 18.4 | 0.02 |
| RF | 81.1 | 12.8 | 1.99 |

than other training modules. This is primarily since decision trees are simpler and faster to design than other ML techniques. Logistic regression's basic optimization approach and linear structure frequently result in short training times. Naïve Bayes algorithms are well known for their simplicity and effectiveness during training. In terms of memory use, we discovered that KNN and decision trees require more memory than other algorithms. This is because decision trees and KNN require a considerable quantity of training dataset information to be stored during model creation. The decision tree algorithm holds information on splits, characteristics, and labels at each node, resulting in increased memory usage. Similarly, KNN loads the whole training dataset into memory, which might be memory-intensive for large datasets. Random Forest and naïve Bayes have higher CPU utilisation compared to other algorithms. This increased CPU utilisation is due to the computational complexity of these techniques during training. Random Forest algorithms typically involve building multiple decision trees and aggregating their predictions, which can require intensive computations, especially for large datasets or a large number of trees in the ensemble. As a result, Random Forest algorithms tend to utilize more CPU resources during the training process. Similarly, naïve Bayes algorithms may require significant computational resources, especially when dealing with high-dimensional data or when estimating probabilities for multiple classes.

From the above analysis, it is clear that tuning the hyperparameter is an important factor in optimizing the proposed framework's performance. Among the trained models, the Decision tree is the most efficient in terms of performance.

## CONCLUSIONS

In this work, we present a FL framework that leverages the exchange of diverse ML models among various entities. Our framework enables the training of ML models on individual datasets while preserving data integrity and privacy. By adopting a federated approach, we address the challenges of data silos and data privacy concerns typically encountered in centralized training scenarios. The proposed framework enables companies to collectively improve the performance of their models by leveraging collective knowledge from distributed datasets, thereby increasing the overall accuracy and effectiveness of ML models used in counterfeit detection and prevention. We have implemented six different ML models to detect counterfeits. We evaluated the performance of the models based on their training and testing accuracy. The models were: decision tree, Random Forest, naive Bayes, logistic regression, SVM, and KNN. Overall, the decision tree model was the most accurate counterfeit detection model. The Random Forest model was also a good choice and its accuracy was very close to the decision tree model. The remaining models have moderate accuracy but can still be useful in some cases. In the proposed framework, blockchain client nodes are referred to as decentralized banking applications. Increasing the number of banking nodes would impact the execution of the composite trained model in the federated server nodes and would also require additional resources in terms of computational power and storage space. The growth in the number of nodes could lead to issues including increased network traffic, latency, and the need for more strong consensus procedures to verify the integrity of transactions across the network. Furthermore, maintaining more client nodes could require stronger security measures to safeguard sensitive financial data and assure regulatory compliance.

The two challenges covered are:

1. The implementation of a federated learning framework helps to reduce counterfeit chances.
2. Different ML models are implemented by each entity or bank to detect counterfeit transactions much faster and accurately, which is the joint effect of all the entities or banks to strengthen their counterfeit-detecting ML algorithms.

In conclusion, this work explored the use of multiple ML models for counterfeit detection and successfully solved the issues of client privacy and collaboration. The FL framework and the deployment of multiple models by different entities or banks demonstrated the effectiveness of these approaches in improving the accuracy and efficiency of counterfeit detection systems. The results obtained from this research have significant implications for enhancing the security and reliability of counterfeit detection mechanisms in various domains. Moreover, the future of counterfeit detection in fintech will likely involve the use of ML and FL. These technologies will help to make fintech more secure and protect customers from counterfeits.

### Funding

This work was supported by the Princess Nourah bint Abdulrahman University Researchers Supporting Project number (PNURSP2024R104), Princess Nourah bint Abdulrahman University, Riyadh, Saudi Arabia. The funders had no role in study design, data collection and analysis, decision to publish, or preparation of the manuscript.

### Grant Disclosures

The following grant information was disclosed by the authors:
Princess Nourah bint Abdulrahman University Researchers Supporting Project: PNURSP2024R104.
Princess Nourah bint Abdulrahman University.

### Competing Interests

The authors declare that they have no competing interests.

### Author Contributions

- Hasnain Rabbani conceived and designed the experiments, performed the experiments, performed the computation work, authored or reviewed drafts of the article, and approved the final draft.
- Muhammad Farrukh Shahid conceived and designed the experiments, performed the experiments, performed the computation work, authored or reviewed drafts of the article, and approved the final draft.
- Tariq Jamil Saifullah Khanzada conceived and designed the experiments, performed the computation work, authored or reviewed drafts of the article, and approved the final draft.
- Shahbaz Siddiqui conceived and designed the experiments, performed the computation work, prepared figures and/or tables, and approved the final draft.
- Mona Mamdouh Jamjoom performed the experiments, analyzed the data, prepared figures and/or tables, authored or reviewed drafts of the article, and approved the final draft.
- Rehab Bahaaddin Ashari performed the experiments, analyzed the data, prepared figures and/or tables, authored or reviewed drafts of the article, and approved the final draft.
- Zahid Ullah analyzed the data, authored or reviewed drafts of the article, and approved the final draft.
- Muhammad Umair Mukati analyzed the data, authored or reviewed drafts of the article, and approved the final draft.
- Mustafa Nooruddin analyzed the data, prepared figures and/or tables, and approved the final draft.

### Data Availability

The data is available at Zenodo: Siddiqui, S. (2024). Enchancing Security [Data set]. Zenodo. https://doi.org/10.5281/zenodo.10444175.

## Supplemental Information

Supplemental information for this article can be found online at http://dx.doi.org/10.7717/peerj-cs.2280#supplemental-information.

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
