# Peer review of "Enhancing security in financial transactions: a novel blockchain-based federated learning framework for detecting counterfeit data in fintech"

_PeerJ Computer Science, doi:10.7717/peerj-cs.2280_

## Round 0.1 · original submission · Major Revisions

Reviewers are suggesting to revise the manuscript on its proposal and its technical aspects. Further, there are concerns about its presentation and articulation as well. Please submit your revised manuscript along with the list of all the corrections carried out.

**Language Note:** PeerJ staff have identified that the English language needs to be improved. When you prepare your next revision, please either (i) have a colleague who is proficient in English and familiar with the subject matter review your manuscript, or (ii) contact a professional editing service to review your manuscript. PeerJ can provide language editing services - you can contact us at [email protected] for pricing (be sure to provide your manuscript number and title). – PeerJ Staff

·

Basic reporting

Based on my review, this manuscript presents a novel blockchain-based federated learning framework for detecting counterfeit data in fintech. Here are my main suggestions:

Overall Quality and Contribution
- The manuscript is well-written with a clear structure conforming to standards. The language is professional and unambiguous. Figures and visuals support the text effectively.

- The proposed framework combining federated learning and blockchain for enhanced security and privacy in fintech is innovative with good potential impact.

- The research aims to address an important real-world problem of increasing counterfeit activities and data breaches in financial services. If published, this work could benefit the fintech industry.

Experimental design

- The framework design linking decentralized clients and central server for collaborative model training is logically presented. The workflow diagrams aid understanding.

- Six common ML models are appropriately implemented for fraud detection. The parameter configurations demonstrate a systematic evaluation.

- However, more details could be provided on the data pre-processing, feature engineering, and model optimization steps.

Validity of the findings

- The results showcase promising model performance, indicating the framework's effectiveness for counterfeit detection. In particular, decision tree and random forest models achieve high testing accuracy.

- Additional validation through more rigorous statistical tests would further strengthen the validity of findings. Comparative analysis against benchmarks could help too.

Additional comments

Suggested Improvements
- Expand the literature review, particularly highlighting latest advances at the intersection of federated learning, blockchain and fintech security.

- Provide more specifics on threat models, trust mechanisms, and privacy-preservation techniques employed.

- Enhance results analysis - include confusion matrices, ROC curves, feature importance metrics for key models.

I hope these suggestions help strengthen this promising research further. Please feel free to let me know if you need any clarification or have additional questions!

·

Basic reporting

Strengths:
1. The paper clearly addresses blockchain-based federated learning for detecting counterfeit data in the fintech industry.

2. The research successfully proposes and implements models for this purpose.

3. Researchers have successfully proposed machine learning-based federated learning models for tackling fraud detection.

4. Researchers have successfully implemented user privacy while implementing a fraud detection system.

5. Researchers have successfully identified machine learning models for detecting counterfeit data in the fintech industry.

Weaknesses:
1. The paper would benefit from a more comprehensive comparison of performance(for accuracy) metrics with existing research in the Literature Review section.
Currently, only one paper is referenced, and expanding this comparison could provide deeper insights into the effectiveness of the proposed research in relation to other studies.

2. While the proposed model maintains a fixed number of input nodes and hidden layers, it would be beneficial to explore the effects of varying these components.
An analysis of different model configurations could enhance understanding of the model's behavior and performance.

3. The implications of varying the number of federated block chain clients should be addressed.
Considering potential impacts and challenges associated with different client configurations would provide a more thorough analysis of the proposed approach.

4. In section "5.3.2 Smart Push and Pull" it is mentioned that "568 Algorithm-5 represents the ’Smart Push and Pull Contract’ that is responsible for forwarding the latest 569 tuned parameters of the trained model to the federated server and pulling the requested aggregated trained 570 model to the decentralized federated server for use by the federated client nodes. "
However, the potential risks of malicious or faulty clients altering model parameters are not adequately addressed.
Exploring safeguards against such threats is crucial to ensure the integrity and privacy of the federated learning system.
Most of the research papers are missing this critical possibility.
A malicious client can alter the model parameters, and it can significantly affect the server-side model parameters, thereby compromising privacy.

Suggestions:
In the context of research, utilizing the Python programming language could offer several advantages over Java.
Python's extensive libraries and frameworks provide robust support for experimentation, prototyping, and implementation, potentially streamlining the research process and facilitating deeper exploration of ideas.

Experimental design

Strengths:
1. The paper clearly addresses blockchain-based federated learning for detecting counterfeit data in the fintech industry.

2. The research successfully proposes and implements models for this purpose.

3. Researchers have successfully proposed machine learning-based federated learning models for tackling fraud detection.

4. Researchers have successfully implemented user privacy while implementing a fraud detection system.

5. Researchers have successfully identified machine learning models for detecting counterfeit data in the fintech industry.

Weaknesses:
1. The paper would benefit from a more comprehensive comparison of performance(for accuracy) metrics with existing research in the Literature Review section.
Currently, only one paper is referenced, and expanding this comparison could provide deeper insights into the effectiveness of the proposed research in relation to other studies.

2. While the proposed model maintains a fixed number of input nodes and hidden layers, it would be beneficial to explore the effects of varying these components.
An analysis of different model configurations could enhance understanding of the model's behavior and performance.

3. The implications of varying the number of federated block chain clients should be addressed.
Considering potential impacts and challenges associated with different client configurations would provide a more thorough analysis of the proposed approach.

4. In section "5.3.2 Smart Push and Pull" it is mentioned that "568 Algorithm-5 represents the ’Smart Push and Pull Contract’ that is responsible for forwarding the latest 569 tuned parameters of the trained model to the federated server and pulling the requested aggregated trained 570 model to the decentralized federated server for use by the federated client nodes. "
However, the potential risks of malicious or faulty clients altering model parameters are not adequately addressed.
Exploring safeguards against such threats is crucial to ensure the integrity and privacy of the federated learning system.
Most of the research papers are missing this critical possibility.
A malicious client can alter the model parameters, and it can significantly affect the server-side model parameters, thereby compromising privacy.

Suggestions:
In the context of research, utilizing the Python programming language could offer several advantages over Java.
Python's extensive libraries and frameworks provide robust support for experimentation, prototyping, and implementation, potentially streamlining the research process and facilitating deeper exploration of ideas.

Validity of the findings

The validity of the findings was well-established through sufficient images and other details provided in the paper. Additionally, the authors were able to supply the source code of the proposed system.

Additional comments

The researchers have successfully proposed and implemented the primary focus of the paper. It has my approval, provided the authors address the minor weaknesses mentioned in the review comments.

---

## Round 0.2 · Minor Revisions

While reviewers are okay with the corrections, they are not convinced with the language and presentation of the paper. Please revise and resubmit after a careful recheck and revision of language and presentation.

·

Basic reporting

A few suggestions/comments on potential further improvements:

It may be helpful to briefly explain the rationale behind choosing OrdinalEncoder over other encoding techniques like one-hot.
For MinMaxScaler, consider specifying the actual value range used (presumed 0-1 based on the text).
Provide a bit more detail on the NearMiss undersampling approach - what variant was used, any tuning of parameters, etc.

Experimental design

A statistical computational analysis was performed to further evaluate the model’s efficiency which looks good.

Validity of the findings

Overall, however, this revised section provides sufficient methodological detail on the data preprocessing steps taken. The authors have adequately addressed my previous comment seeking more elaboration in this area. Well done in strengthening this part of the manuscript.

Additional comments

n/a

·

Basic reporting

"INTRODUCTION" section lacks smooth reading due to multiple spelling mistakes, most of which are due to missing spaces between words.

For example, on page 3, corrections are needed for "counterfeitcan". Space is missing.

"counterfeitinvolves", "counterfeitwhere", "counterfeitinvolves" are some other instances.

Only a few are mentioned here.
Update the section accordingly.

Experimental design

Researchers were able to update the paper with the review suggestions mentioned.

Validity of the findings

no comment

Additional comments

Researchers were able to update the paper with the review suggestions mentioned.

---

## Round 0.3 · Minor Revisions

Your revision did not satisfy the reviewer. Please carefully consider the comments and revise the work accordingly in this round.

·

Basic reporting

I accept the final review and it looks good

Experimental design

I accept the final review and it looks good

Validity of the findings

I accept the final review and it looks good

Additional comments

n/a

·

Basic reporting

Strengths:
1. The authors were able to incorporate blockchain-based federated learning (FL) in this research.

2. The authors implemented various supervised machine learning models and compared various output metrics.

3. The authors successfully implemented blockchain-based FL in fintech.


Weaknesses:
1. The authors did not explain which features from the input dataset collected from Kaggle were used in this study. They mentioned the use of some features for the OrdinalEncoder but did not specify the other features in the input dataset.

2. The authors did not clearly mention various model parameter updating strategies, such as Model Weight Update Using Average and Model Weight Update Using Federated Averaging (FedAvg). This is a core aspect of the FL process.

3. The authors did not delve deeply into the concepts of FL. Numerous research papers connect blockchain and FL and use various supervised learning techniques. The authors failed to advance the research beyond this basic level.

Experimental design

Strengths:
1. The authors were able to incorporate blockchain-based federated learning (FL) in this research.

2. The authors implemented various supervised machine learning models and compared various output metrics.

3. The authors successfully implemented blockchain-based FL in fintech.


Weaknesses:
1. The authors did not explain which features from the input dataset collected from Kaggle were used in this study. They mentioned the use of some features for the OrdinalEncoder but did not specify the other features in the input dataset.

2. The authors did not clearly mention various model parameter updating strategies, such as Model Weight Update Using Average and Model Weight Update Using Federated Averaging (FedAvg). This is a core aspect of the FL process.

3. The authors did not delve deeply into the concepts of FL. Numerous research papers connect blockchain and FL and use various supervised learning techniques. The authors failed to advance the research beyond this basic level.

Validity of the findings

no comment

Additional comments

Strengths:
1. The authors were able to incorporate blockchain-based federated learning (FL) in this research.

2. The authors implemented various supervised machine learning models and compared various output metrics.

3. The authors successfully implemented blockchain-based FL in fintech.


Weaknesses:
1. The authors did not explain which features from the input dataset collected from Kaggle were used in this study. They mentioned the use of some features for the OrdinalEncoder but did not specify the other features in the input dataset.

2. The authors did not clearly mention various model parameter updating strategies, such as Model Weight Update Using Average and Model Weight Update Using Federated Averaging (FedAvg). This is a core aspect of the FL process.

3. The authors did not delve deeply into the concepts of FL. Numerous research papers connect blockchain and FL and use various supervised learning techniques. The authors failed to advance the research beyond this basic level.

---

## Round 0.4 · accepted · Accept

After checking the article, I believe the authors have revised the work accordingly. I can be accepted currently. Congrats!